# Identifying Patients at Risk of Acute Kidney Injury among Patients Receiving Immune Checkpoint Inhibitors: A Machine Learning Approach

**DOI:** 10.3390/diagnostics12123157

**Published:** 2022-12-14

**Authors:** Xiang Yu, Rilige Wu, Yuwei Ji, Mengjie Huang, Zhe Feng

**Affiliations:** 1Department of Nephrology, The First Medical Center of Chinese PLA General Hospital, Chinese PLA Institute of Nephrology, Beijing Key Laboratory of Kidney Disease, State Key Laboratory of Kidney Diseases, National Clinical Research Center for Kidney Diseases, Chinese PLA General Hospital, Beijing 100853, China; 2Medical Big Data Research Center, Chinese PLA General Hospital, Beijing 100853, China

**Keywords:** immune checkpoint inhibitors, acute kidney injury, machine learning, predictive models

## Abstract

**Background**: The benefits of immune checkpoint inhibitors (ICPis) in the treatment of patients with malignancies emerged recently, but immune-related adverse events (IRAEs), including acute kidney injury (AKI), cannot be ignored. The present study established and validated an ICPi-AKI prediction model based on machine learning algorithms to achieve early prediction of AKI events and timely intervention adjustment. **Methods**: We performed a retrospective study based on data from the First Medical Center of the PLA General Hospital. Patients with malignancy who received at least one dose of ICPi between January 2014 and December 2019 were included in the study. The characteristics of available variables were included after case review, and the baseline characteristics and clinical data of ICPi AKI and non-AKI patients were compared. After variable preprocessing, eight machine learning algorithms were used to construct a full variable availability model. Variable simplification models were constructed after screening important variables using the random forest recursive feature elimination method, and the performance of different machine learning methods and two types of modeling strategies were evaluated using multiple indicators. **Results**: Among the 1616 patients receiving checkpoint inhibitors, the overall incidence of AKI was 6.9% during the total follow-up time. Sixty-eight patients were associated with ICPi treatment after chart review, primarily in AKI stage 1 (70.5%), with a median time from first ICPi administration to AKI of 12.7 (IQR 2 to 56) weeks. The demographic characteristics, comorbidities, and proportions of malignancy types were similar between the ICPi-AKI and non-AKI groups, but there were significant differences in multiple characteristics, such as concomitant medications and laboratory test indicators. For model performance evaluation and comparison, the AUC values of all 38 variable availability models ranged from 0.7204–0.8241, and the AUC values of the simplicity model constructed using 16 significant variables ranged from 0.7528–0.8315. The neural networks model (NNs) and support vector machine (SVM) model had the best performance in the two types of modeling strategies, respectively; however, there was no significant difference in model performance comparison (*p* > 0.05). In addition, compared with the full variable availability model, the performance of the variable simplicity model was slightly improved. We also found that concomitant medications contributed more to the model prediction performance by screening the optimal feature combination. **Conclusion**: We successfully developed a machine learning-based ICPi-AKI prediction model and validated the best prediction performance of each machine model. It is reasonable to believe that clinical decision models driven by artificial intelligence can improve AKI prediction in patients with malignancies treated with ICPi. These models can be used to assist clinicians in the early identification of patients at high risk of AKI, support effective prevention and intervention, and ultimately improve the overall benefit of antitumor therapy in the target population.

## 1. Introduction

Immune checkpoint inhibitors (ICPis) are a novel and promising anticancer therapy. The primary mechanism of action of these humanized monoclonal antibodies is inhibition of downstream immune pathways and a reprogramming of adaptive immunity to recognize and destroy tumor cells via host immune cells [1]. The FDA approved the first ICPi against cytotoxic T-lymphocyte antigen 4 (CTLA-4) in 2011, and subsequent antibodies against programmed cell death 1 (PD1) and programmed death ligand 1 (PDL-1) have been gradually developed and used in the clinic while new drugs have entered clinical trials.

ICPi therapy has transformed many malignancies from a “death sentence” to a chronic disease. However, the benefits of this therapy are not perfect, and extensive T-cell suppression also leads to autoimmune side effects, which are termed immune-related adverse events (IRAEs). These IRAEs most commonly affect the skin, gastrointestinal tract, liver, and endocrine system but may involve any organ system, including the kidneys [1,2]. The incidence of acute kidney injury (AKI) in single ICPi treatment is approximately 2%, and it is up to 4.5% when a combination of ICPis is used [3]. Some studies suggest that the failure to recover renal function in AKI patients is an independent predictor of death [4], and AKI may affect the overall therapeutic effect in patients due to termination of ICPi medication.

A spectrum of kidney IRAEs has been described in case reports and smaller series, including acute interstitial nephritis (AIN), acute tubular necrosis (ATN), and, less commonly, glomerular disease [4,5,6]. Recent clinical studies analyzed the pathogenic factors of ICPi-AKI, and the risk factors include anemia, diuretics, nonsteroidal anti-inflammatory drugs (NSAIDs), and proton pump inhibitors (PPIs) [3,4,7].

Due to the impact of ICPi-AKI on the treatment of patients with malignancies, the early identification of target treatment populations with a high risk of AKI is highly important. The construction of predictive modeling is promising, but such studies are lacking. The classical logistic regression model is sensitive to the multicollinearity of independent variables, which makes the model easy to be underfitted and far from accurate. Artificial intelligence-based machine learning is booming and creating a technological revolution, especially in the healthcare industry. Compared to traditional statistical analysis methods, such as logistic regression, machine learning more effectively identifies complex relationships between diseases and variables, classifies variables using specific criteria, makes predictions based on baseline features, and identifies objects with similar patterns. Therefore, these models help in disease identification and inference, assist in medical decision-making, and improve the quality of treatment [7]. Current research on machine learning modeling is rich.; Ebiaredoh-Mienye et al. [8,9] developed effective prediction models for chronic kidney disease CKD, which is easy to ignore due to a lack of obvious early symptoms. Superior classification performance with over 98% accuracy was achieved in both studies. The positive effects of the improved sparse autoencoder (SAE) network and information-gain-based feature selection technique on various classifiers were verified. Research on AKI is also relatively rich. For example, Koyner et al. (2018) confirmed the predictive efficacy of machine learning models for patients with acute renal injury before creatinine changes [10]. Zhang et al. (2019) successfully demonstrated the potential of machine learning methods to distinguish volume-responsive and volume-unresponsive AKI [11]. Tomasev’s team and Google Deepmind, which is based on artificial neural network algorithms, have pushed the research on AKI machine learning prediction models to another high [12]. In Section 2 and Section 3 of this paper, we introduce the modeling methods and model results, respectively, and in Section 5, we discuss and summarize this research. We fully drew on technical experience from previous studies to develop an ICPI-AKI prediction model using machine learning algorithms for predicting the occurrence of long-term AKI events in the target population based on variable characteristics during the first ICPi treatment.

## 2. Materials and Methods

### 2.1. Data Sources and Study Population

We performed a single-center retrospective study based on data from the First Medical Center of the PLA General Hospital, which is a large military-affiliated hospital in Beijing with more than 200 nursing units that treat nearly 200,000 patients annually. Patients with malignancy who received at least one dose of ICPis between January 2014 and December 2019 and whose malignancy diagnosis was based on the International Classification of Diseases code (ICD-9/10) were included in the study. Patients aged <18 years, renal transplant patients, end-stage patients on maintenance hemodialysis, and patients with incomplete follow-up data were excluded from the study. This study was a retrospective case collection. The real IDs of all study subjects were privacy marked before analysis, no special interventions were performed, and there was no impact on patient safety and health. The ethics committees of PLA General Hospital approved the study (S2020-356).

### 2.2. Study Design

The study was divided into five steps. **First**, we relied on the big data center of the PLA General Hospital for data identification and extraction of patients treated with ICPis, wrote structured query language (SQL) scripts based on diagnostic criteria, and screened AKI cases in the Oracle database 10 g. We used the 2012 Kidney Disease: Improving Global Outcomes (KDIGO) definition of AKI as the source of major screening criteria [8]. AKI was defined as an increase in serum creatinine (SCr) by 26.5 μmol/L within 48 h or a 50% increase from baseline during the follow-up period [9]. The baseline SCr was the most recent value within three months prior to the first ICPi dose, and the peak SCr level within three months after AKI onset was used to determine AKI stage according to the following KDIGO criteria: stage 1, the absolute SCr level increased to 1.5–1.9 times the baseline level; stage 2, the SCr level increased to 2.0–2.9 times the baseline level; and stage 3, the absolute SCr level increased to more than 3 times the baseline level or greater than 353.6 µmol/L, or renal replacement therapy (RRT) was initiated [13]. Case follow-up periods were equivalent to the duration of ICPi treatment until cessation due to tumor progression, IRAEs requiring treatment adjustment and/or discontinuation of treatment, or death events. Urine volume was not analyzed in this study due to the uncertainty of urine volume measurement and the instability of available data.

**Second**, a novel definition and classification system for ICPi-AKI, proposed by Gupta et al. [14], was used to distinguish ICPi-AKI from other AKI etiologies (non-ICPi-AKI). This classification acknowledges several gradations of diagnostic uncertainty in the absence of kidney biopsy. **Definite ICPi-AKI** was a diagnosis of ICPI-AKI confirmed by renal biopsy after clinical review of risk factors. **Probable ICPi-AKI** was assigned when three criteria were met after clinical review of risk factors: (1) SCr elevation ≥1.5 times the baseline value on at least two consecutive values or the need for RRT; (2) absence of an alternative plausible cause; and (3) at least one of the three additional criteria of sterile pyuria, eosinophilia, or recent or concomitant non-kidney IRAE. **Possible ICPI-AKI** was an increase in SCr ≥50% or the need for RRT, and the AKI was not readily attributable to alternative causes. Two nephrologists charted all AKI cases for review, and a third nephrologist resolved diagnostic differences.

**Third**, the variables were selected according to clinical experience and considering significant variables in previous ICPi-AKI risk factor studies. Highly qualified nephrologists on the research team reviewed the variables, and AI engineers performed technical confirmation. We restricted our research to structured data because the processing of unstructured data (e.g., clinical notes). Although these data provide additional diagnostic and laboratory information, it relies on natural language processing (NLP) technology, which could directly affect our performance comparisons between different models. The 38 variables included demographic information, comorbidities, laboratory tests, ICPis used, concomitant medications, and malignancy type, and all variables were included in at least three other peer-reviewed studies. Medical records information was reviewed and extracted such that the time of exposure was the date of first dosing with ICPis, comorbidities were diagnosed before the first ICPi treatment, laboratory test values were the most recent values within one week before the first dose of ICPis (except SCr), and concomitant medications were a combination of drugs administered within one month before and three weeks after the first dose of ICPis, including drugs that alter renal hemodynamics and medications that induce allergic interstitial nephritis (AIN). End-stage renal disease (ESRD) was defined as an estimated glomerular filtration rate (eGFR) <15 mL-min^−1^—(1.73 m^2^)^−1^, and chronic kidney disease (CKD) was defined as eGFR <60 mL-min^−1^—(1.73 m^2^)^−1^. eGFR was calculated using the eGFR-EPI formula [15], and the data were based on the time automatically calibrated in the system.

**Fourth**, processing of screening variables and outlier processing of continuous variables in the database were performed to remove extreme values beyond the 1–99th percentile and exclude interference of extreme values due to entry errors (e.g., age, BMI, eGFR, and laboratory test values). Categorical variables were converted to binary variables based on exposure, i.e., “yes” or “no” (e.g., ICI drugs, concomitant medications, and comorbidities). For missing data, variables with missing values >20% were discarded. Otherwise, missing values in the data were filled via random forest interpolation (missForest).

**Fifth**, multiple machine learning methods were used to construct the model. This was because the ratio of the number of non-AKI to ICPi-AKI cases was approximately 23:1, and the data were severely unbalanced. The downsampling method was used to process the data, and a portion of the samples were randomly selected from the non-AKI cases to ensure that the ratio of the two groups was approximately 3:1. Seventy percent of the data after downsampling processing was randomly selected as the training group, and 30% was used as the testing group. The training group data were used for model training, and the testing group data were used for model testing (Figure 1). In the modeling process, the parameters of the models were continuously adjusted using fivefold cross-validation to reduce the chances of overfitting. In addition, we introduced the concepts of a prediction time window and event time window in the modeling process to achieve information matching between ICPi-AKI and non-AKI patients. The range of the prediction event window was from the day before the first dose of ICPi to 21 days after the treatment, and the range of the event time window was from 21 days after the first dose of ICPi treatment to the end of follow-up. All variables were included in the prediction time frame, and the time of the first creatinine value meeting the KDIGO criteria in the event window was referred to as the AKI event time (Figure 2).

By incorporating all 38 feature variables, eight supervised learning methods were used to construct the full variable availability model: logistic regression (LR), decision tree (DT), random forest (RF), support vector machine (SVM), extreme gradient boosting (Xgboost), adaptive boosting (Adaboost), naïve Bayes (NBs), and neural networks (NNs). To improve the usefulness of the model, we incorporated a reasonable number of independent variables. Therefore, the random forest recursive feature elimination (RF-RFE) method was used to determine the optimal feature combination variables, and the optimal solution was used as a parameter to determine the feature variables that would be included in the model. The variable simplicity model was established in this manner.

### 2.3. Model Evaluation

The area under the receiver operating characteristic curve (AUROC), accuracy, recall, Matthews’ correlation coefficient (MCC), precision, and F1 value were used as model efficacy evaluation metrics, and the confusion matrix of the optimal prediction model was also shown. The AUC value of the model is significant for predictive modeling and decision-making. Therefore, AUC was the main metric used in model selection and final report.

### 2.4. Statistical Analysis

Continuous variables meeting the conditions of normal distribution are presented as the mean (SD), and one-way analysis of variance (ANOVA) was used for between-group comparisons. Continuous variables with nonnormal distribution are presented as the median (IQR), and the Mann–Whitney U test was used for between-group comparisons. Categorical variables are presented as n (%), and Pearson’s chi-squared test was used for between-group comparisons. Delong test was used for model performance comparisons. A two-sided *p* value of <0.05 was considered significant in all statistical analyses. The R language data analysis software R-Studio (Version 7.2, RStudio Institute Inc., Boston, MA, USA) was used for data reading and analytical modeling.

## 3. Results

A total of 1616 of the 1649 patients who received at least one dose of ICPis were included in the study cohort. All cases were reviewed for AKI events according to the KDIGO criteria. A total of 111 (6.9%) patients met the AKI diagnosis criteria during the total follow-up time (Figure 3). As shown in Table 1, 68 patients (4.2%) were classified as having an ICPi-AKI diagnosis according to the cause differentiation and diagnostic certainty classification of AKI using the Gupta method. Forty-three of these patients (63.2%) were male, and the median age was 57.00 years (IQR 49.00 to 64.25). The most common cancers included were lung cancer (26.5%), hepatobiliary cancer (26.5%), and gastrointestinal tract cancer (20.6%). The most common comorbidities were diabetes (22.1%), hypertension (35.3%), and liver disease (17.6%). The most common types of ICPis were nabolumab (55.9%) and pablizumab (39.7%). The most common concomitant medications were PPIs (95.6%) and NSAIDs (82.4%).

As shown in Table 2, there were no patients with a definite diagnosis of ICPi-AKI because kidney biopsy was not recorded in all patients. Twenty-one (30.9%) and forty-seven (69.1%) patients met the criteria of probable and possible ICPi-AKI, respectively. Forty-eight (70.5%) patients had stage 1 AKI, fifteen (22.1%) patients had stage 2 AKI, and five (7.4%) patients had stage 3 AKI. The median time from the first ICPi administration to ICPi-AKI was 12.7 (IQR 2 to 56) weeks. Nine patients (13.2%) received a nephrology consultation. Urinalysis was performed in 55 patients (80.9%). Eleven (16.2%) patients had leukocyturia that met the diagnostic criteria, nineteen patients (27.9%) had microscopic hematuria, and thirty-five patients (51.5%) had positive albumin. Twelve (17.6%) patients developed extrarenal IRAEs, including immune-associated pneumonia (5/7.4%), immune-associated hepatitis (3/4.4%), immune-associated pleural effusions (2/2.9%), immune-associated enteritis (1/1.5%), and immune-associated fever (1/1.5%), and were treated with glucocorticoids.

Forty-three patients were classified with a non-ICPi-AKI diagnosis. Thirty-five (81.4%) of these diagnoses were likely related to a prerenal cause, such as hemodynamics, four (9.3%) diagnoses were attributed to postrenal obstruction (one bladder epithelial carcinoma, one ovarian plasmacytoma, one rectal adenocarcinoma, and one colon adenocarcinoma), three (7.0%) diagnoses were associated with contrast agents (two hepatocellular carcinoma embolization PCI procedures in one case), and one (2.3%) diagnosis was associated with nephrectomy for nephrocalcinoma.

### 3.1. Comparison of Clinical Characteristics between Groups

Table 1 shows the comparison of clinical characteristics between the ICPi-AKI group and the non-AKI group. The results showed that age, sex, and BMI characteristics were relatively similar between groups. Patients in both groups were more likely to have diabetes and hypertension, and most patients in both groups received ICPis as treatments for lung cancer, hepatobiliary cancer, and gastrointestinal tract cancer. Compared to the non-AKI group, patients in the ICPi-AKI group were more likely to have received nivolumab, antibiotics, diuretics, NSAIDs, and PPIs (*p* < 0.05), and had lower mean HB, ALB, baseline SCr, and HCT (*p* < 0.05) and higher median NE, baseline eGFR, and D-dimer (*p* < 0.05). As shown in Table 3, except for sindilizumab and D-dimer, there were no significant differences between patients in the training and testing groups in clinical characteristics (*p* > 0.05).

### 3.2. Significant Variable Screening

To increase the clinical applicability of the model, we performed model feature screening using the RF-RFE method. Figure 4 shows that the model with 16 features had the best diagnostic accuracy in cross-validation, and the model classification error rate continued to increase with the inclusion of more features. Cross-validation was performed on different combinations of features based on random forest with the learner itself unchanged, by calculating the sum of decision coefficients to ultimately obtain the importance of different features for the results and retaining the best combination of features. The final combination of 16 significant variables that we identified included one demographic feature (age), one ICPi (nivolumab), six concomitant medications (NSAIDS, antibiotics, PPI, ACEI/ARB, diuretics, and chemotherapeutic agents), four laboratory test indicators (ALb, Hb, eGFR, and D-dimer), and four malignancy types (hepatobiliary cancer, genitourinary cancer, gastrointestinal tract cancer, and lung cancer). These variables were closely related to AKI events and should guide clinicians to pay more attention.

### 3.3. Model Performance Evaluation

Based on the training group, models were constructed using machine learning methods, and performance was evaluated in the validation group. Table 4 shows the results of multiple model performance tests based on the independent validation group, which was trained and evaluated using all 38 variables. The AUC values ranged from 0.7204–0.8241 for all eight models, and the NNs model showed the best overall performance with an AUC of 0.8167. The accuracy was 0.7703, the recall was 0.7, the precision was 0.56, the F1 value was 0.6222, and the MCC was 0.4660. The SVM model had similar performance, and the DT, XGBoost, NBs, and AdaBoost models had intermediate performance. Although the LR model had the worst performance, its AUC value and accuracy were more reliable, however, there is no significant difference in prediction performance between different models (*p* > 0.05) (Appendix A). The receiver operating characteristic curves are shown in Figure 5a. The confusion matrix for the NNs model is also showed in Figure 6a.

Table 5 shows that based on the same independent validation group, the evaluation results of multiple models constructed with 16 significant variables screened using the RFE method revealed that all 8 models had an AUC range of 0.7528–0.8315. The SVM model showed optimal performance with an AUC of 0.8315, an accuracy of 0.7568, a recall rate of 0.6, a precision of 0.6667, and an F1 value of 0.6077. The variable simplicity model achieved a balance between minimizing the number of features and maximizing the performance improvement relative to the models constructed with all 38 variables, with seven of the models showing a performance improvement of 0.0009–0.05 (excluded NNs); however, there is no significant difference in prediction performance between different models (*p* > 0.05) (Appendix A). Figure 5b shows the receiver operating characteristic curves for multiple models in the validation group. Figure 6b shows the confusion matrix for the SVM model.

## 4. Discussion

Based on a local dataset, we successfully developed and validated a machine learning prediction model to identify high-risk patients with ICPi-AKI. The overall performance of the simplified model with 16 variables was slightly better than the full variable availability model, and known risk factors, such as nephrotoxic drugs, were predictors of ICPi-AKI and contributed more to the model prediction performance.

Because of the extrarenal clearance mechanism of ICPis, its toxic effects on the kidney have been neglected and underestimated for many years. The exact incidence of ICPi-AKI is not certain because of the heterogeneity of the definition of ICPi-AKI. For example, the mild symptoms of stage 1 AKI are considered in some studies, but case determination is performed with stage 2 or higher AKI stages. The etiology of ICPi-AKI is not clearly differentiated in many studies, which results in the existence of false positive cases. A lack of follow-up information due to the delayed onset of ICPi-AKI is also one of the reasons. The present study used the KDIGO criteria for case screening and found that the burden of AKI in patients treated with ICPis was considerable, with an all-cause AKI incidence of 6.9% (111/1615). The high incidence of AKI in this population may reflect multiple causes of renal dysfunction, including ICPi-related nephrotoxicity and other forms of impairment that are inherent to patients receiving anticancer therapy. A chart review of AKI cases was also performed using the Gupta method [14], and the incidence of ICPi-AKI was calculated to be 4.2%. This method allowed for a detailed review of all AKI patients to exclude non-ICPI-AKI cases and avoid the misclassification of cases when renal biopsy was refused. According to the results of routine urine tests and extrarenal IRAEs, more AKI cases without reasonable etiology were recorded as probable or possible cases. The incidence results of our study cohort were between the results of Koks et al. [16] and Cortazar et al. [4]. The etiological classification method in the former study was the same as our study, with a calculated incidence of 4.7%. The latter study included 3695 patients treated with ICPis in phase 2 and 3 clinical trials for meta-analysis, and the estimated incidence of ICPi-AKI was 2.2%. However, our incidence was significantly lower than Meraz-Muñoz et al. [17], who reported an ICPi-AKI incidence of 9.7%. Analysis of the reasons for the discrepancy suggests that 14% of AKI cases with a duration of less than 72 h were excluded from the Meraz-Muñoz et al. [17] study. However, our study made no distinction between persistent AKI because most patients had mild AKI (stage 1), and the interval between SCr measurements was significantly longer than 72 h in all cases.

Increasing application of machine learning in predicting AKI (or prognosis) has been seen in past research reports, and the technology has been modified and implemented for different susceptible populations, including patients in general wards, patients with severe disease, patients with pancreatitis or sepsis, and patients with SGLT-1 hypoglycemic agents, using different modeling strategies. However, there are no studies on ICPi-treated malignancy patients with a high risk of AKI. Minimizing any organ damage is necessary due to the complex disease environment in oncology patients, which affects their quality of survival and treatment benefit. Riding the fast train of artificial intelligence in the big data era increases the possibility to predict and prevent AKI events. We focused on information at the time the patient received the first dose of ICPi treatment and included only information on malignancy diagnosis, prevalent comorbidities, concomitant medication orders, and partial test indicators that were easily available from the electronic health record (EHR) to optimize the various evaluation metrics based on the chosen thresholds. The SVM model showed the best performance, with outstanding advantages in several effectiveness evaluation metrics, such as AUC, accuracy, recall, precision and F1 value, which were significantly higher than the LR models. These results are similar to previous AKI prediction model studies. For example, Sun et al. [18] used the open-source database MIMIC-III as the basis for an AKI prediction model for intensive care unit inpatients using structured and unstructured feature configurations. The SVM model achieved a competitive AUC value of 0.83. Qu et al. [19] used the SVM model for predicting AKI associated with pancreatitis and achieved an AUC value of 0.86 and a specificity of 0.85. However, models, such as XGboost, outperformed the SVM in the latter study. One similarity between that study and our study is that both studies had relatively small positive and total sample sizes, which demonstrates the computational modeling advantages of the SVM, such as greater applicability to small sample size data, insensitivity to outliers, excellent generalization ability, and the identification and effective use of key variables. Our study suggests that the performance metrics of the 16 variables modeled after filtering with the RFE method increased slightly, which ensures the stronger clinical usability of the model. Our results indirectly demonstrated that sex, BMI, comorbidity, certain types of ICPis and most clinical test indictors were not always reliable predictive biomarkers for ICPi-AKI. Therefore, our model should better distinguish ICPi patients with a high risk of AKI, regardless of the type of ICPi drug they are receiving (except nivolumab) and whether they have a previous underlying disease or equivalent underlying physiological status, which was reflected by test indictors, such as lymphocyte count and baseline SCr.

The optimal combination of variables in this study was screened according to the RFE method, and the results suggested that the predictive contribution of AKI events was closely related primarily to primary cancer type and medication orders, where drugs included PPIs, ACEIs/ARBs, NSAIDs, diuretics, antibiotics, chemotherapy, and nivolumab. PPIs increase the risk of AKI. The possible pathogenic mechanism involves PPI activation of effector T cells, especially when T cells become sensitized with the use of ICPi, and this effect gradually increases the risk of AKI [20,21]. Therefore, it may be beneficial to replace H_2_ receptor blockers in patients with a potential risk of ICPi-AKI. No direct association between NSAIDs and ICPi-AKI risk has been observed. Previous series hypothesized that receipt of ICPis modified immune tolerance to these drugs, and these agents should be discontinued after ICPi-AIN and potentially prior to rechallenge with ICPis [4,6]. The findings on the association of diuretics and ACEI/ARB drugs with ICPi-AKI are also controversial. It has been suggested that, except for immune effects that are similar to PPI, the pathogenic mechanisms of drugs in both of these categories may be related to prerenal factors [16]. The multifactorial analysis of several studies has not confirmed antibiotics as an independent risk factor for AKI events, but antibiotics are believed to increase the incidence of AKI [1,22]. However, systemic prophylactic antibiotic use greatly reduces the progression-free survival and overall survival of patients treated with ICPis, possibly by mechanisms related to antibiotic-induced alterations in the intestinal flora and effects on the body’s immunity [23]. Nivolumab is one of the most widely used ICPis in clinical treatment, and it was considered harmless to the kidney in early studies [24,25]. However, its correlation with AKI events has been gradually emphasized [26]. Nivolumab-related AKI is more likely to occur within 6–12 months after treatment and is more sensitive to glucocorticoid therapy [27,28]. However, the exact pathogenic mechanism is poorly understood.

The type of malignancy addressed in the present study had a high value for AKI prediction, but the correlation between malignancy type and ICPi-AKI was inconclusive in previous studies. Gynecological malignancies were previously associated with a 3.91-fold increased risk of AKI in Koks et al. [16]. This correlation may be related to postrenal obstruction caused by the tumor itself, but ureteral obstruction and retroperitoneal fibrosis indirectly caused by radiotherapy are also important causative factors [29]. Several previous studies confirmed that the overall incidence of AKI in patients with NSCLC treated with nivolumab ranged from 1.9% to 3.4% [27,30,31]. The incidence of AKI was less than 1% in patients treated with pembrolizumab monotherapy [32,33]. Lung cancer accounted for nearly 40% of the patients in our cohort. Although a direct correlation between lung cancer and ICPi-AKI could not be confirmed in other parallel studies, it remained a strong predictor and deserves clinical attention, especially when ICPis are combined with other chemotherapeutic agents. Cancer of the liver and biliary system is one of the common refractory cancers, and it has a strong immune-mediated pathogenesis that allows ICPi to exert significant antitumor effects and become an effective alternative after sorafenib treatment failure. Pembrolizumab, camrelizumab and tislelizumab are also recommended as second-line regimens for hepatocellular carcinoma in several guidelines [34,35,36]. Gastrointestinal tract cancer is a global health problem, and prospective research results support the use of ICPis in the third-line treatment of advanced gastric cancer [37,38,39]. Nivolumab and pembrolizumab have been approved for third-line treatment in the United States and Japan. Nephrotoxicity has not been explicitly stated in relevant studies related to these two types of malignancy, and it has not been confirmed as an independent risk factor for ICPi-AKI in subsequent studies. Therefore, the present study adds to the conclusion of its importance in predicting the risk or grade of AKI for some patients with malignant tumors treated with ICPis, even if the predictive factors are not equivalent to risk factors.

The study used the Gupta method to distinguish the cases of ICPi-AKI to avoid the incorrect classification of cases without renal biopsy evidence in previous studies. Compared to previous studies, our study had a relatively large sample size, which helped us more effectively demonstrate and supplement the results of previous studies, especially risk factors. Our research cohort was rich in malignant tumor diagnosis, which increases the applicability of the model. Our study also has some limitations. First, the proportion of ICPI-AKI patients was significantly lower than non-AKI patients, which affects the generalization power and reliability of the model even after downsampling to balance the sample. Second, there were no kidney biopsy cases in the study cohort, and etiological classification according to the Gupta method could only confirm a probable or possible diagnosis. However, the method reduced selection bias to some extent compared to other studies. Third, the high proportion of patients with urinary erythrocytosis seen in this study cohort did not exclude ICPi-derived glomerular injury, but the lack of renal biopsy records did not allow further verification. Fourth, this study was a retrospective cohort with few data on the use of PD-L1 and the clinical practice of ICPi combination therapy. Therefore, we could not provide data and conclusions to demonstrate the specific effect of combination therapy on ICPi-AKI. Sixth, there is no difference in the performance of different models in this study, which may be related to the small sample size. In the future, it is planned to continue to collect cases, expand the sample size, further optimize the model parameters, and improve the model performance.

## 5. Conclusions

The present study successfully developed an ICPI-AKI prediction model based on a machine learning method for the first time. This model only needs a small number of clinically available variables for early prediction and risk assessment of ICPI-AKI events to prevent clinicians from ignoring key AKI clues and adopt timely adjustment of medical advice and effective preventive measures. While ensuring the consistency of ICPi treatment, it can avoid the adverse effects of kidney puncture and hormone administration in some people and ultimately improve the overall therapeutic benefits of malignant tumor patients and the quality of life of the target population. Future studies will develop an electronic tool based on this model and integrate it with physician workstations to optimize the prevention and treatment management of ICPI-AKI using the alert function to combine effective care decisions. We will perform prospective validation in other medical centers to continuously optimize the performance of the model.

## Figures and Tables

**Figure 1 diagnostics-12-03157-f001:**
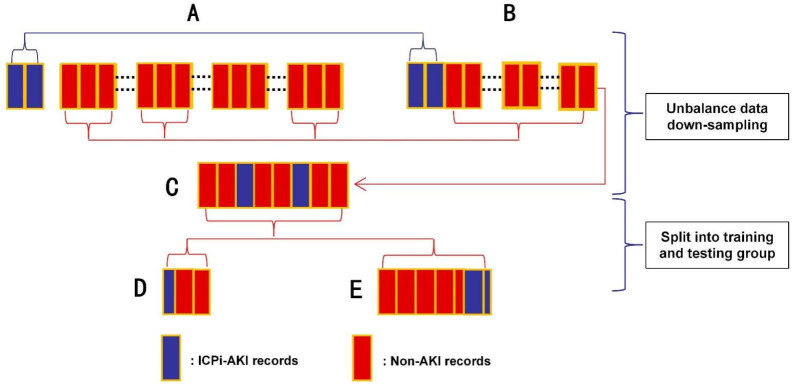
Handling and splitting of imbalanced data.Note: (**A**) original patient cohort included in the study; (**B**,**C**) patient cohort after down-sampling; (**D**) testing group; (**E**) training group.

**Figure 2 diagnostics-12-03157-f002:**
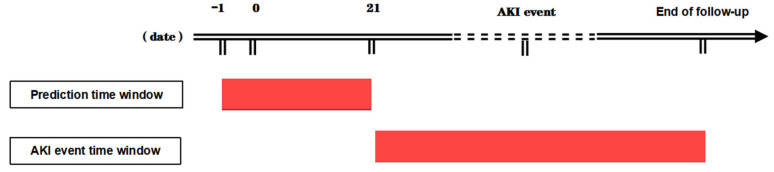
Time window setting of the prediction model.

**Figure 3 diagnostics-12-03157-f003:**
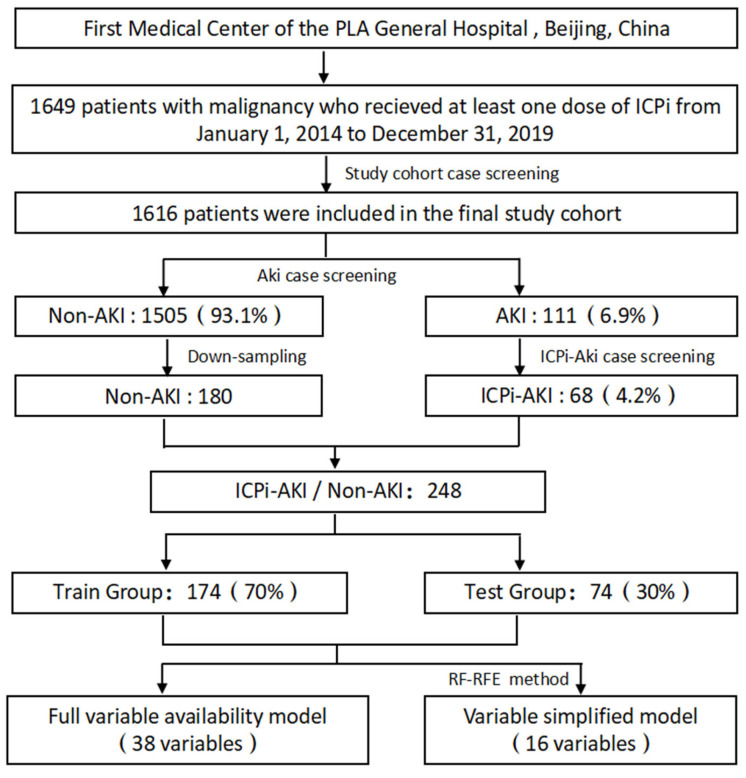
Flowchart of the data preparation process pertaining to ICPi-AKI and Non-AKI patients. Abbreviations: ICPi, immune checkpoint inhibitor; AKI, acute kidney injury.

**Figure 4 diagnostics-12-03157-f004:**
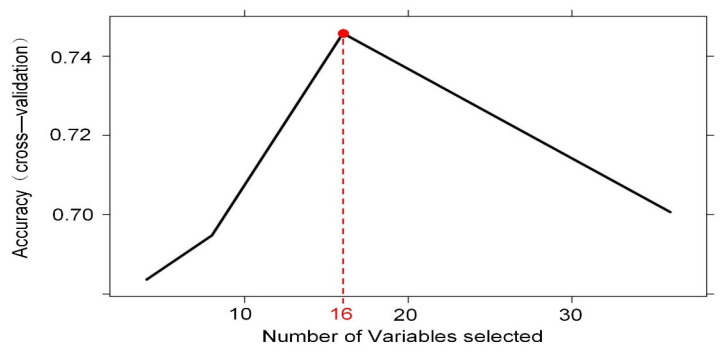
Visualization of optimal model variable combination quantity based on RF-RFE.

**Figure 5 diagnostics-12-03157-f005:**
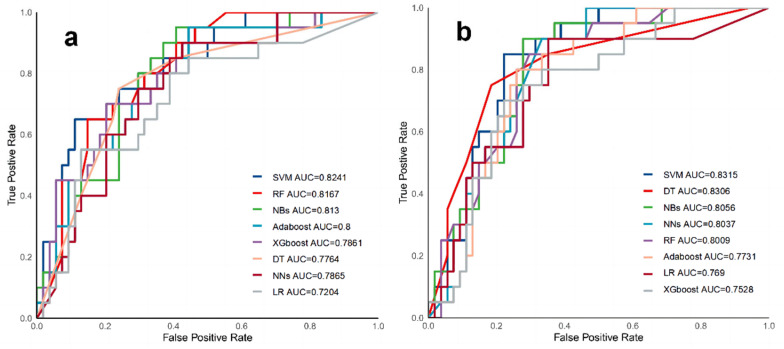
The receiver operating characteristic (ROC) curves of 8 different models. **Note**: (**a**) Models based on all variables. (**b**) Models based on important variables. **Abbreviations**: SVM, support vector machine; RF, random forest; DT, decision tree; LR, logistic regression; NNs, neural networks; Adaboost, adaptive boosting; XGboost, extreme gradient boosting; NBs, naive Bayes.

**Figure 6 diagnostics-12-03157-f006:**
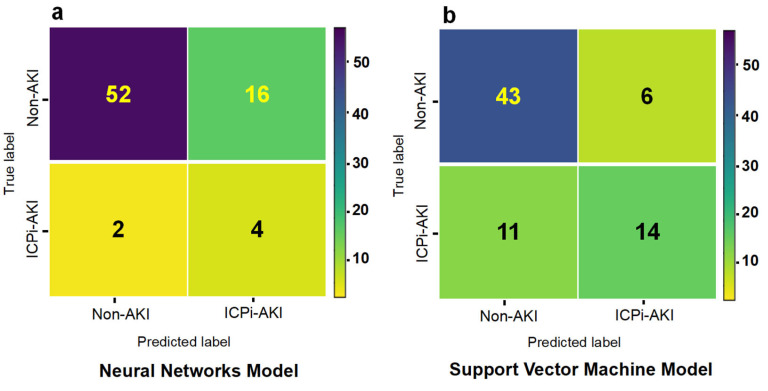
Confusion matrix of the Neural Networks model and Support Vector Machine model. **Note**: Using all 38 variables to construct the model, the Neural Network model has the best performance, and the confusion matrix is shown in the (**a**); Using 16 significant variables to construct the model, the Support Vector Machine model has the best performance, and the confusion matrix is shown in the (**b**).

**Table 1 diagnostics-12-03157-t001:** Characteristics of the ICPi-AKI and non-AKI groups.

Variables	Overall (n = 248)	Non-AKI (n = 180)	ICPi-AKI (n = 68)	*p* Value
Male (%)	181 (73.0)	138 (76.7)	43 (63.2)	
Age, median (IRQ)	59.50 (51.00, 67.00)	60.00 (52.75, 67.00)	57.00 (49.00, 64.25)	0.053
BMI, median (IRQ)	23.10 (20.38, 25.40)	23.10 (20.50, 25.40)	22.75 (19.90, 25.45)	0.521
Malignancy type (%)				0.16
Mammary cancer	2 (0.8)	2 (1.1)	0 (0.0)	
Colorectum cancer	15 (6.0)	11 (6.1)	4 (5.9)	
Gastrointestinal tract cancer	42 (16.9)	28 (15.6)	14 (20.6)	
Genitourinary cancer	17 (6.9)	15 (8.3)	2 (2.9)	
Hepatobiliary cancer	52 (21.0)	34 (18.9)	18 (26.5)	
Lung cancer	91 (36.7)	73 (40.6)	18 (26.5)	
Melanoma	2 (0.8)	1 (0.6)	1 (1.5)	
Other	27 (10.9)	16 (8.9)	11 (16.2)	
Concomitant medications				
ACEI/ARB (%)	32 (12.9)	30 (16.7)	2 (2.9)	0.008
Antibiotic (%)	9 (3.6)	3 (1.7)	6 (8.8)	0.021
Diuretic (%)	75 (30.2)	46 (25.6)	29 (42.6)	0.014
NSAIDS (%)	125 (50.4)	69 (38.3)	56 (82.4)	<0.001
Chemotherapy (%)	68 (27.4)	46 (25.6)	22 (32.4)	0.362
PPI (%)	186 (75.0)	121 (67.2)	65 (95.6)	<0.001
Comorbidity				
Diabetes (%)	43 (17.3)	28 (15.6)	15 (22.1)	0.308
Hypertension (%)	68 (27.4)	44 (24.4)	24 (35.3)	0.121
Coronary heart disease (%)	22 (8.9)	19 (10.6)	3 (4.4)	0.205
Cerebrovascular (%)	11 (4.4)	9 (5.0)	2 (2.9)	0.721
Liver disease (%)	38 (15.3)	26 (14.4)	12 (17.6)	0.669
ICPi type				
Nivolumab (%)	99 (39.9)	61 (33.9)	38 (55.9)	0.003
Pembrolizumab (%)	86 (34.7)	59 (32.8)	27 (39.7)	0.383
Ipilimumab (%)	9 (3.6)	5 (2.8)	4 (5.9)	0.432
Toripalimab (%)	23 (9.3)	23 (12.8)	0 (0.0)	0.004
Sintilimab (%)	40 (16.1)	34 (18.9)	6 (8.8)	0.084
Camrelizumab (%)	3 (1.2)	3 (1.7)	0 (0.0)	0.674
Atezolizumab (%)	5 (2.0)	4 (2.2)	1 (1.5)	1
Laboratory test indicators				
HB, mean (SD)	116.56 (21.87)	120.04 (21.88)	107.37 (19.11)	<0.001
WBC, median (IQR)	6.22 (4.50, 7.81)	6.22 (4.48, 8.01)	6.32 (4.59, 7.60)	0.946
PLT, median (IQR)	199.00 (147.75, 259.50)	206.50 (152.75, 259.50)	183.50 (138.00, 257.25)	0.186
NE, median (IQR)	0.70 (0.62, 0.78)	0.69 (0.61, 0.77)	0.74 (0.66, 0.79)	0.038
LYM, median (IQR)	0.19 (0.14, 0.26)	0.20 (0.14, 0.27)	0.18 (0.13, 0.24)	0.093
ALB, mean (SD)	37.75 (4.61)	38.41 (4.41)	35.99 (4.70)	<0.001
SCR, median (IQR)	68.50 (55.35, 81.85)	70.30 (57.58, 82.78)	60.45 (50.55, 76.42)	0.006
ALT, median (IQR)	17.00 (11.20, 25.77)	16.30 (11.20, 24.90)	18.35 (11.15, 33.73)	0.459
AST, median (IQR)	19.30 (14.88, 29.65)	19.40 (14.45, 27.85)	19.00 (15.33, 34.67)	0.423
eGFR, median (IQR)	96.90 (83.23, 107.25)	95.43 (80.63, 104.96)	103.62 (89.14, 111.47)	0.006
LDH, median (IQR)	181.60 (148.07, 256.58)	180.45 (150.05, 249.10)	183.95 (145.67, 285.07)	0.598
D-DIMER, median (IQR)	1.04 (0.52, 2.67)	0.93 (0.44, 2.43)	1.36 (0.81, 3.80)	0.003
HCT, mean (SD)	0.34 (0.06)	0.35 (0.06)	0.32 (0.06)	<0.001

Abbreviations: HB, hemoglobin; WBC, white blood cell, PLT, platelet; NE, neutrophil; LYM, lymphocyte; ALB, albumin; SCR, serum creatinine; ALT, alanine aminotransferase; AST, aspartate aminotransferase; eGFR, estimated glomerular filtration rate; LDH, lactate dehydrogenase; D-DIMER, D2 polymers; HCT, hematocrit; ACEI, angiotensin converting enzyme inhibitors; ARB, angiotensin receptor blocker; NSAIDS, nonsteroidal anti-inflammatory drugs; PPI, proton pump inhibitors; BMI, body mass index; ICPi, immune checkpoint inhibitor.

**Table 2 diagnostics-12-03157-t002:** Clinical data of patients with ICPi-AKI.

Patients with ICPi-AKI	N = 68 (4.2% of all Patients)
AKI stage (%)	
Stage 1	48 (70.5)
Stage 2	15 (22.1)
Stage 3	5 (7.4)
Gradations of diagnostic uncertainty	
Definite ICPi-AKI	0
Probable ICPi-AKI	21 (30.9)
Possible ICPi-AKI	47 (69.1)
Urinalysis results	55 (80.9)
Leukocyturia	11 (16.2)
Microscopic hematuria	19 (27.9)
Albuminuria	35 (51.5)
Extrarenal IRAEs	12 (17.6)
Immune associated pneumonia	5 (7.4)
Immune associated hepatitis	3 (4.4)
Immune related pleural effusion	2(2.9)
Immune associated enteritis	1 (1.5)
Immune associated fever	1 (1.5)

**Table 3 diagnostics-12-03157-t003:** Characteristics between the training and test groups.

Variables	Overall (n = 248)	Training Group (n = 174)	Testing Group (n = 74)	*p* Value
ICPi-AKI (%)	68 (27.4)	48 (27.6)	20 (27.0)	
Male (%)	181 (73.0)	124 (71.3)	57 (77.0)	0.436
Age, median (IRQ)	59.50 [51.00, 67.00]	60.00 [52.00, 67.00]	59.00 [51.00, 66.00]	0.673
BMI, median (IRQ)	23.10 [20.38, 25.40]	23.20 [20.33, 25.28]	22.55 [20.40, 25.67]	0.823
Malignancy type (%)				0.779
mammary cancer	2 (0.8)	1 (0.6)	1 (1.4)	
Colorectum cancer	15 (6.0)	12 (6.9)	3 (4.1)	
Gastrointestinal tract cancer	42 (16.9)	30 (17.2)	12 (16.2)	
Genitourinary cancer	17 (6.9)	11 (6.3)	6 (8.1)	
Hepatobiliary cancer	52 (21.0)	36 (20.7)	16 (21.6)	
Lung cancer	91 (36.7)	66 (37.9)	25 (33.8)	
Melanoma	2 (0.8)	2 (1.1)	0 (0.0)	
Other	27 (10.9)	16 (9.2)	11 (14.9)	
Medication (%)				
ACEI/ARB	32 (12.9)	18 (10.3)	14 (18.9)	0.102
Antibiotic	9 (3.6)	6 (3.4)	3 (4.1)	1
Diuretic	75 (30.2)	51 (29.3)	24 (32.4)	0.735
NSAIDS	125 (50.4)	86 (49.4)	39 (52.7)	0.739
Chemotherapy	68 (27.4)	53 (30.5)	15 (20.3)	0.136
PPI	186 (75.0)	136 (78.2)	50 (67.6)	0.109
Comorbidity (%)				
Diabetes	43 (17.3)	30 (17.2)	13 (17.6)	1
Hypertension	68 (27.4)	47 (27.0)	21 (28.4)	0.948
Coronary heart disease	22 (8.9)	16 (9.2)	6 (8.1)	0.975
Cerebrovascular	11 (4.4)	7 (4.0)	4 (5.4)	0.883
Liver disease	38 (15.3)	27 (15.5)	11 (14.9)	1
Checkpoint inhibitor type				
Nivolumab	99 (39.9)	70 (40.2)	29 (39.2)	0.991
Pembrolizumab	86 (34.7)	58 (33.3)	28 (37.8)	0.592
Ipilimumab	9 (3.6)	6 (3.4)	3 (4.1)	1
Toripalimab	23 (9.3)	14 (8.0)	9 (12.2)	0.433
Sintilimab	40 (16.1)	34 (19.5)	6 (8.1)	0.04
Camrelizumab	3 (1.2)	2 (1.1)	1 (1.4)	1
Atezolizumab	5 (2.0)	2 (1.1)	3 (4.1)	0.32
Laboratory test indicators				
HB, mean (SD)	116.56 (21.87)	117.33 (22.16)	114.76 (21.20)	0.397
WBC, median (IQR)	6.22 [4.50, 7.81]	6.08 [4.42, 7.79]	6.74 [4.79, 7.87]	0.145
PLT, median (IQR)	199.00 [147.75, 259.50]	188.00 [147.25, 248.75]	218.00 [149.50, 277.75]	0.097
NE, median (IQR)	0.70 [0.62, 0.78]	0.70 [0.62, 0.77]	0.71 [0.64, 0.78]	0.478
LYM, median (IQR)	0.19 [0.14, 0.26]	0.20 [0.14, 0.27]	0.17 [0.13, 0.23]	0.067
ALB, mean (SD)	37.75 (4.61)	37.94 (4.71)	37.29 (4.36)	0.309
SCr, median (IQR)	68.50 [55.35, 81.85]	67.15 [55.12, 81.15]	71.70 [56.90, 82.38]	0.241
ALT, median (IQR)	17.00 [11.20, 25.77]	17.30 [11.35, 26.10]	16.00 [11.20, 24.77]	0.407
AST, median (IQR)	19.30 [14.88, 29.65]	18.80 [14.30, 29.20]	20.55 [15.53, 30.50]	0.504
eGFR, median (IQR)	96.90 [83.23, 107.25]	97.90 [84.88, 108.09]	93.86 [83.16, 104.74]	0.174
LDH, median (IQR)	181.60 [148.07, 256.58]	179.60 [147.55, 246.17]	190.70 [151.12, 278.98]	0.295
D-DIMER, median (IQR)	1.04 [0.52, 2.67]	0.98 [0.44, 2.46]	1.15 [0.68, 3.66]	0.041
HCT, mean (SD)	0.34 (0.06)	0.35 (0.06)	0.34 (0.06)	0.677

Abbreviations: HB, hemoglobin; WBC, white blood cell, PLT, platelet; NE, neutrophil; LYM, lymphocyte; ALB, albumin; SCr, serum creatinine; ALT, alanine aminotransferase; AST, aspartate aminotransferase; eGFR, estimated glomerular filtration rate; LDH, lactate dehydrogenase; D-DIMER, D2 polymers; HCT, hematocrit; ACEI, angiotensin converting enzyme inhibitors; ARB, angiotensin receptor blocker; NSAIDS, nonsteroidal anti-inflammatory drugs; PPI, proton pump inhibitors; BMI, body mass index; ICPi, immune checkpoint inhibitor.

**Table 4 diagnostics-12-03157-t004:** The performance of different models based on all variables.

Model	Accuracy	AUROC	F1-Score	Recall	Precision	MCC
Support Vector Machine (SVM) with radial kernel	0.7297	0.8093	NA	0	NA	-
Support Vector Machine (SVM) with sigmoid kernel	0.7568	0.7787	0.5	0.45	0.5625	0.3456
Support Vector Machine (SVM) with polynomial kernel	0.7297	0.813	NA	0	NA	-
Decision Tree (DT)	0.7568	0.7764	0.625	0.75	0.5357	0.4663
Random Forest (RF)	0.7432	0.8241	0.2963	0.2	0.5714	0.2192
logistic Regression (LR)	0.7703	0.7204	0.5405	0.5	0.5882	0.3910
Neural Networks (NNs)	0.7703	0.8167	0.6222	0.7	0.56	0.4660
Adaptive Boosting (Adaboost)	0.7568	0.8	0.5714	0.6	0.5455	0.4030
Extreme Gradient Boosting (XGboost)	0.6892	0.7685	0.4651	0.5	0.4348	0.2488
Naïve Bayes (NBs)	0.7297	0.7861	0.5833	0.7	0.5	0.4036

**Table 5 diagnostics-12-03157-t005:** The performance of different models based on important variables.

Model	Accuracy	AUROC	F1-Score	Recall	Precision	MCC
Support Vector Machine (SVM) with radial kernel	0.7568	0.8315	0.60769	0.6	0.66667	0.2651
Support Vector Machine (SVM) with sigmoid kernel	0.7432	0.8287	0.42424	0.35	0.53846	0.2788
Support Vector Machine (SVM) with polynomial kernel	0.7297	0.8213	NA	0	NA	-
Decision Tree (DT)	0.7973	0.8056	0.6341	0.65	0.619	0.4944
Random Forest (RF)	0.7297	0.8306	0.375	0.3	0.5	0.2276
logistic Regression (LR)	0.7568	0.7528	0.5263	0.5	0.5556	0.3642
Neural Networks (NNs)	0.6622	0.769	0.5098	0.65	0.4194	0.2850
Adaptive Boosting (Adaboost)	0.7162	0.8009	0.4324	0.4	0.4706	0.2463
Extreme Gradient Boosting (XGboost)	0.7297	0.7713	0.4737	0.45	0.5	0.2933
Naïve Bayes (NBs)	0.7432	0.8037	0.5778	0.65	0.52	0.4017

## Data Availability

Not applicable.

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
