# Peer review of "Identifying Patients at Risk of Acute Kidney Injury among Patients Receiving Immune Checkpoint Inhibitors: A Machine Learning Approach"

_diagnostics, 2022, doi:10.3390/diagnostics12123157_

Round 1
Reviewer 1 Report
This paper employs several machine learning algorithms coupled with clinical characteristics data to induce ICPi-AKI prediction model, used to predict unseen examples. The paper is well-explained. However, the following comments need to be addressed:
-Need to report performance results on the whole dataset using 5-fold cross-validation; Report combined confusion matrix; and
Include other performance measures such as Matthews correlation coefficient. Also, report the standard deviation
-Report p-values to assess the statistical difference between the models and to inspect if results obtained by SVM are significant or not
Author Response
Please see the attachmentdd

Reviewer 2 Report
The authors did a good job by using the AI/ML approach in Identifying Patients at Risk of Acute Kidney Injury among patients Receiving Immune Checkpoint Inhibitors.
Secondly, there are other recent investigations that have extensively done good work in this regard for instance;
A Machine Learning Method with Filter-Based Feature Selection for Improved Detection of Chronic Kidney Disease” Bioengineering 2022, vol. 9, no. 8, 350; https://doi.org/10.3390/bioengineering9080350, Switzerland.
An Interpretable Machine Learning Approach for Hepatitis B Diagnosis”. Applied Sciences. 2022; 12(21):11127. https://doi.org/10.3390/app122111127.
“Integrating Enhanced Sparse Autoencoder Based Artificial Neural Network Technique and SoftMax Regression for Medical Diagnosis. Electronics Journal,2020, 9(11), 1963; https://doi.org/10.3390/electronics9111963 Switzerland.
And many more you can search out.
Reviewer 3 Report
In the study, researchers identified patients at risk of acute kidney injury among patients taking immune checkpoint inhibitors using a machine learning approach. The study seems appropriate, but the following issues need to be eliminated.
1) The Introduction is very missing. This part needs some improvement. The problem is not defined in this section. Why artificial intelligence approach was used should be mentioned. In addition to these, the highlights of the study should be emphasized before the last paragraph of the Introduction.
2) The organization of the study should be given in the last paragraph of the Introduction.
3) A literature review was not performed. At least five different current articles in this field or similar fields should be scanned and added to the study.
4) Is the dataset used in the study a public dataset? If so, it would be nice if the access link to the data is provided on the article.
5) The flow chart of the study is not given. The profile of the study is given in Figure 3. It is not very understandable. Therefore, a flow chart or a graphical abstract of the study should be given.
6) The performances of machine learning algorithms were determined by evaluation criteria such as accuracy and F1-score. In addition to these, the complex matrix is one of the most important evaluation matrices used in health fields. Researchers should at least include the complexity matrix of the most successful classification algorithm in the article.
7) The advantages and disadvantages of the study should be added at the end of the Discussion section. What kind of solutions have been produced to what kind of problems? Or the limitations of the methods used should be mentioned.
8) The Conclusion seems very incomplete. In this section, the contribution of the study to the literature should be mentioned and the study should be summarized. If there are studies planned to be done in the future, these should also be mentioned briefly.
9) The similarity rate of the study was 20%. This must be at least 15%. The similarity report of the study is given as an appendix.

Round 2
Reviewer 1 Report
Authors addressed comments.
Minor comments:
-in line 37 at the abstract,
CHANGE
however, there is no significant difference
TO
however, there was no significant difference
-Figure 3 caption (in line 230) is not appropriate,
CHANGE
flow chart
TO (possibly something like)
Flowchart of the data preparation process pertaining to ICPi-AKI and non-AKI patients
-In Figure 6 (confusion matrix), need to change 0 and 1 to ICPi-AKI and non-AKI according to your encoding. That is important for the reader
Reviewer 3 Report
The authors have complied with all the revisions given. However, what I meant by the organization of the article was something like this:
In the second section, methods and materials are mentioned. In the third section, the results are examined. Something like that. You don't have to, of course, but it will help increase the readability of the article.
